# A Forecasting Benchmark of Large-Scale Neural Populations during Task-Driven Behavior

**Giuliano Costa & Petru Manescu**
Department of Computer Science
University College London
London WC1E 6BT, United Kingdom
{ucabgdc}@ucl.ac.uk

## Abstract

While substantial advances have been made in decoding behavioral or stimulus variables from 2-photon calcium imaging data—the forward problem of predicting future neural activity from past population dynamics remains underexplored. We leverage a novel, large-scale calcium imaging dataset from the International Brain Laboratory, where mice perform a complex decision-making task, to systematically evaluate neural forecasting. Using 11,393 neurons recorded simultaneously for over an hour, we applied standard forecasting techniques to assess the predictability of this system. Through experiments examining population scaling, prediction horizon, and cortical state transitions, we find that forecasting performance exhibits diminishing returns with population size and degrades substantially when models encounter unseen brain states. Simple univariate models remain competitive with sophisticated multivariate architectures, suggesting current approaches fail to exploit higher-order population codes. This work establishes a single-session benchmark for neural population forecasting on a large-scale mesoscope recording from behaviorally-engaged mice.

## 1 Introduction

The mammalian brain is constantly making predictions. From anticipating sensory consequences of motor actions to modeling environment changes, neural circuits continuously generate forecasts that guide perception and behavior. There has been substantial progress on decoding behavior and stimulus from large populations of neurons. But forecasting future population dynamics from past activity remains largely unexplored. The ability to accurately forecast neural activity has direct implications for computational theories of brain function: if models can learn to predict neural dynamics, they may capture the same computational principles the brain itself employs.

Neuroscience is prospering from large-scale recordings and advancements in recording techniques, and with it a surge in foundation models (Azabou et al., 2024; Le & Shlizerman, 2022; Ye et al., 2023; Zhang et al., 2024), yet systematic investigation of neural forecasting have only recently emerged. Two concurrent works have begun to address this gap: ZAPBench (Lueckmann et al., 2025), a single-session benchmark using whole-brain lightsheet microscopy from zebrafish, and POCO (Duan et al., 2025), a foundation model that leverages multi-session, multi-species calcium recordings for population-conditioned prediction. These works represent important first steps, yet fundamental questions remain about how population scale, cortical state, and preprocessing methodology affect the predictability of mammalian neural dynamics.

**A Novel Large-Scale Dataset.** To address these questions, we leverage an upcoming and unpublished calcium imaging dataset from the International Brain Laboratory (IBL), where mice are recorded during a standardized decision-making task (International Brain Laboratory et al., 2025). This dataset offers critical advantages over existing alternatives. **#1 scale:** 11,393 neurons are recorded simultaneously across multiple cortical regions; two orders of magnitude larger than the 10-40 neurons typical in Allen Brain Observatory two-photon sessions (de Vries et al., 2020). **#2**

**behavioral engagement:** unlike passive viewing paradigms where neural dynamics may reflect metabolically conservative states, the IBL task-driven protocol ensures cortical circuits remain in a desynchronized state optimized for sensory discrimination (Harris & Thiele, 2011; Poulet & Crochet, 2019). **#3 cortical state transitions:** each session includes both an active behavioral task and a passive stimulus portion, enabling direct comparison of forecasting performance across fundamentally different brain states within the same neural population.

**Research Gaps.** Key unresolved questions motivate our investigation. *Population scaling*: while standard time-series benchmarks contain hundreds of variates, neural recordings capture thousands of simultaneous signals. It remains unclear whether multivariate models can effectively leverage cross-neuron dependencies at this scale, or whether performance simply saturates. *Cortical state generalization*: the mammalian brain exhibits shifts markedly different spiking activity depending on attentive state (Harris & Thiele, 2011), providing a unique opportunity to evaluate whether learned representations transfer across desynchronized (active) and synchronized (passive) task protocols. *Architectural requirements*: given that simple linear models often match transformer performance on standard forecasting benchmarks (Zeng et al., 2022), it is unclear whether neural data's unique statistical structure demands more sophisticated architectures.

**Contributions.** We address these gaps through systematic evaluation across population scales (70, 700, 7000 neurons), prediction horizons ($P$=8, 16, 32 timesteps), and cortical state transitions (active task vs. passive viewing). Our key findings include: (1) population scaling exhibits diminishing returns, with a 10-fold increase from 70 to 700 neurons yielding an average 108% improvement versus only 22% from 700 to 7000 neurons; (2) all models degrade when tested on an unseen passive cortical state; (3) univariate models remain surprisingly competitive with multivariate architectures, suggesting multivariate models are unable to capture population codes as seen in Figure 2. This work establishes a single-session benchmark on large-scale mesoscope data that will serve the community as the IBL dataset becomes publicly available.

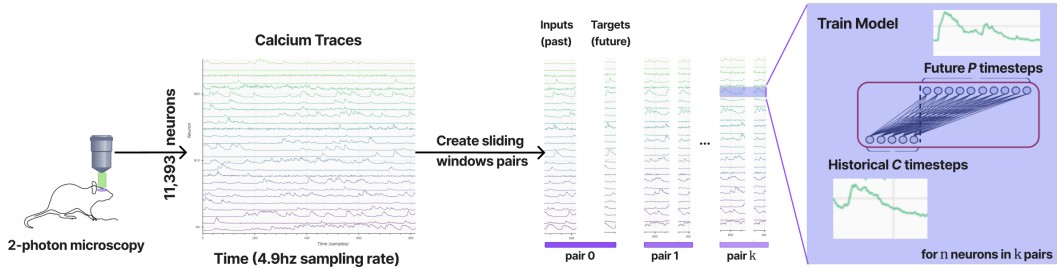

Figure 1: **Our approach to population forecasting.** We leverage the large number of neurons recorded simultaneously during 2-photon calcium imaging to train several forecasting models.

## 2 METHODS

### 2.1 DATASET

Data was collected from a single 75.2 minute recording session of a transgenic mouse expressing GCaMP6s in excitatory neurons performing the International Brain Laboratory (IBL) standardized decision-making task (International Brain Laboratory et al., 2025). Neural activity was recorded using 2-photon calcium imaging across 8 fields-of-view, capturing 11,393 neurons distributed across 11 cortical regions at a sampling rate of 4.9 Hz. The session consisted of 61.1 minutes of active task behavior (521 trials) followed by 14.1 minutes of passive imaging. ROI extraction was performed using Suite2p (Pachitariu et al., 2017). Full recording details are provided in Appendix C.

### 2.2 PREPROCESSING

Our preprocessing pipeline transforms raw 2-photon calcium imaging data into a normalized activity matrix suitable for time-series forecasting. The pipeline consists of two main stages: calculating

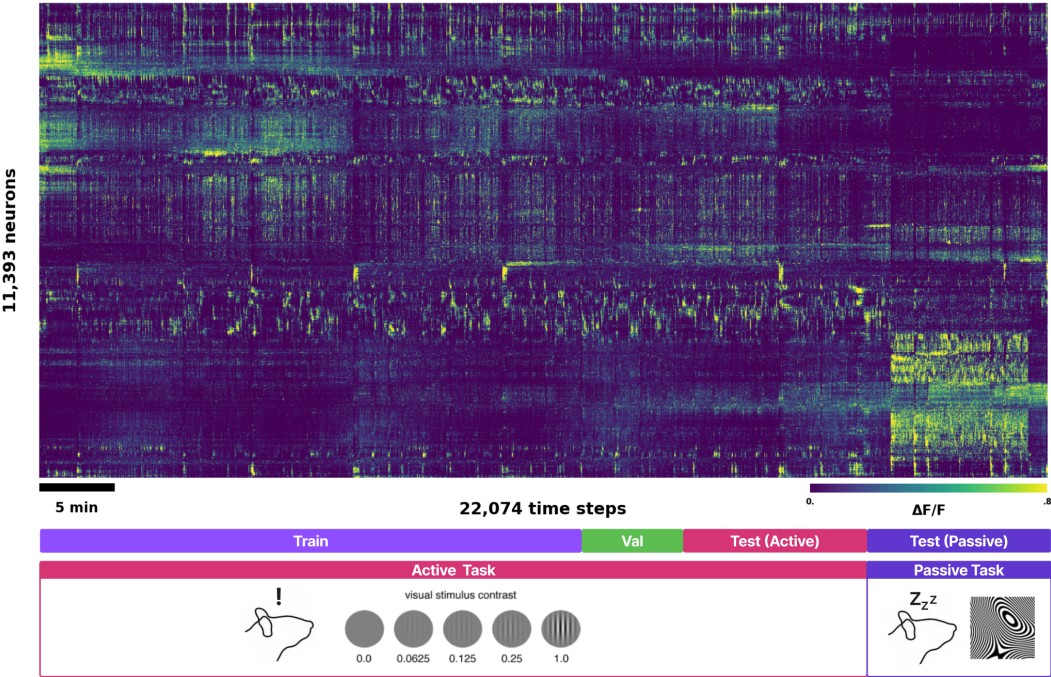

Figure 2: **Rastermap of the activity traces used in the cortical state change experiment.** Color represents normalized $\Delta F/F$ activity, with brighter colors representing higher neuron activity at any one pixel. Neurons are ordered by similarity using Rastermap (Stringer et al., 2025). Data partitioning is shown as a colored bar: train (purple), validation (green), active test (red), passive test (dark purple). For the cortical state experiment, all 11,393 neurons are used with 38 min training, 11 min validation, and two size-matched 13 min test sets — one held-out active segment and the **passive** out-of-distribution period. For the population scaling experiments, subsets of 70–7000 neurons were partitioned 70/20/10 within the **active** portion only.

the change in fluorescence over baseline ($\Delta F/F$) and applying a robust normalization scheme to standardize the dynamic range across neurons.

## 2.3 NEUROPIL CORRECTION AND DELTA F/F CALCULATION

Raw fluorescence signal for each region of interest (ROI), denoted as $F_{\text{raw}}$, is contaminated by background fluorescence from out-of-focus neuropil. We correct for this using the standard Suite2p method (Pachitariu et al., 2017). The corrected fluorescence, $F_{\text{corr}}$, for each neuron at each time point $t$ is computed by subtracting a fraction of the corresponding neuropil signal $F_{\text{neuropil}}$:

$$F_{\text{corr}}(t) = F_{\text{raw}}(t) - \alpha \cdot F_{\text{neuropil}}(t) \tag{1}$$

where the neuropil coefficient $\alpha$ is set to the standard value of 0.7.

Following neuropil correction, we calculate the $\Delta F/F$, which represents the relative change in fluorescence from a baseline level, $F_0$. The baseline $F_0$ for each neuron is estimated as the 10th percentile of its corrected fluorescence trace, $F_{\text{corr}}$. A minimum baseline value of 1.0 is enforced to prevent division by near-zero values. The $\Delta F/F$ is then given by:

$$\frac{\Delta F}{F}(t) = \frac{F_{\text{corr}}(t) - F_0}{F_0} \tag{2}$$

## 2.4 Robust Per-Neuron Normalization

To ensure that the activity of different neurons is comparable and to stabilize model training, we apply a robust normalization method to each neuron's $\Delta F/F$ trace independently. This method is designed to be insensitive to outliers and extreme values characteristic of calcium imaging data.

For each neuron's time series, we define a robust dynamic range using the 0.5th ($q_{0.5}$) and 99.5th ($q_{99.5}$) percentiles of its activity. The data is first scaled to this range:

$$X_{\text{scaled}}(t) = \frac{\frac{\Delta F}{F}(t) - q_{0.5}}{\max(q_{99.5} - q_{0.5}, 0.1)} \tag{3}$$

After scaling, the data is clipped to a range of [-0.25, 1.5] to remove extreme outliers while preserving some negative fluctuations, which can be physiologically meaningful. We did not apply any temporal smoothing, to preserve the raw temporal dynamics of the signal. The final activity matrix, $X_{\text{final}}$, is therefore the normalized and clipped data.

## 2.5 Models

### 2.5.1 Baselines

**Naive Repeat**

A common baseline in forecasting is simply to repeat the last value in the context window $C$. (Duan et al., 2025; Zeng et al., 2022; Wu et al., 2021) We term this $f_{\text{repeat}}$, where $P$ is the number of prediction steps, $t$ is the beginning of the prediction window, and $x_{t-1}$ the last value in the context window.

$$f_{\text{repeat}}(x^{(i)}_{t-C:t}) = [x^{(i)}_{t-1}, x^{(i)}_{t-1}, ..., x^{(i)}_{t-1}] \quad \text{(repeat for } P \text{ steps)} \tag{4}$$

Despite being extremely simple, the slow dynamics of calcium traces make $f_{\text{repeat}}$ a strong baseline, especially when $P$ is low.

**Mean Repeat**

We also implement a mean repeat baseline that computes the mean of the historical window and repeats this value for all $P$. This baseline $f_{\text{mean}}$ calculates the average of the context window $x^{(i)}_{t-C:t}$ and repeats it for $P$ steps:

$$f_{\text{mean}}(x^{(i)}_{t-C:t}) = [\bar{x}^{(i)}, \bar{x}^{(i)}, ..., \bar{x}^{(i)}] \text{ where } \bar{x}^{(i)} = \frac{1}{C} \sum_{j=t-C}^{t-1} x^{(i)}_j \quad \text{(repeat for } P \text{ steps)} \tag{5}$$

This baseline serves to assess whether models are merely predicting the trivial case of the historical mean, ensuring that more sophisticated models demonstrate genuine predictive capability. It generally performs better than $f_{\text{repeat}}$ and is notably consistent across context and prediction window lengths.

### 2.5.2 Univariate

Univariate models process each neuron independently, effectively fitting separate models for each variate. Each neuron's future activity is predicted solely from its own temporal history. These models cannot capture population-level dynamics or inter-neuron dependencies, but serve as important baselines for assessing whether multivariate complexity provides genuine forecasting improvements.

**Linear**

Following (Zeng et al., 2022), who demonstrated that simple linear models can outperform complex transformer architectures on multiple forecasting benchmarks, we adapted their linear forecasting model. The model, $f_{\text{Linear}}$, applies a single linear transformation to map the historical context window to a future prediction. The function is defined as:

$$\hat{x}^{(i)}_{t:t+P} = f_{\text{Linear}}(x^{(i)}_{t-C:t}; W, b) = W x^{(i)}_{t-C:t} + b \tag{6}$$

where $x_{t-C:t}^{(i)} \in \mathbb{R}^C$ is the context window for neuron $i$, and the output $\hat{x}_{t:t+P}^{(i)} \in \mathbb{R}^P$ is the corresponding prediction. The model uses a single linear transformation with shared parameters, a weight matrix $W \in \mathbb{R}^{P \times C}$ and a bias vector $b \in \mathbb{R}^P$, to map the $C$ input timesteps to $P$ output predictions across all neurons.

**DLinear** DLinear modifies Linear by incorporating a decomposition scheme from Autoformer (Wu et al., 2021), also proposed by (Zeng et al., 2022). Rather than applying a single linear transformation, DLinear first decomposes each input time series $x_{t-C:t}^{(i)}$ into trend and seasonal components using a moving average filter with kernel size $2\lfloor C/4 \rfloor + 1$. Two separate linear layers then independently forecast each component, with the final prediction computed as their sum. Trend components may reflect underlying metabolic processes or behavioral state changes, while seasonal components might capture rapid spiking.

### 2.5.3 MULTIVARIATE

Multivariate models jointly process all neurons simultaneously, enabling cross-neuron information flow during prediction. These architectures can model population-level interactions where one neuron's future activity depends on other neurons' historical patterns.

**TSMixer**

TSMixer is a multivariate MLP-based architecture that processes temporal and neuronal dimensions in an alternating fashion (Chen et al., 2023). Each TSMixer block applies an MLP along the time dimension (time-mixing) to capture temporal dependencies, followed by transposition and two MLPs applied along the neuron dimension (feature-mixing) to model cross-neuron interactions. This design enables the model to learn both temporal patterns within individual neurons and population-level dynamics across the neural ensemble.

**Transformer & Informer**

We evaluate both the standard Transformer (Vaswani et al., 2017) with full $O(L^2)$ attention and Informer (Zhou et al., 2021), which reduces complexity to $O(L \log L)$ via sparse attention and sequence distilling. Despite their multivariate nature, both perform poorly regardless of population size.

**POCO (Population-Conditioned Forecaster)**

POCO (Duan et al., 2025) combines a univariate MLP forecaster with a population encoder that conditions predictions on global brain state using Feature-wise Linear Modulation (FiLM). The model operates through two components: (1) a univariate MLP that processes each neuron's historical context independently, and (2) a PerceiverIO-based population encoder that extracts conditioning parameters $(\gamma, \beta)$ from the entire neural population. The complete model modulates the MLP's hidden activations:

$$f_{\text{POCO}}(x_{t-C:t}) = W_{\text{out}} \left[ \text{ReLU}(W_{\text{in}} x_{t-C:t} + b_{\text{in}}) \odot \gamma + \beta \right] + b_{\text{out}} \tag{7}$$

where $\odot$ denotes element-wise multiplication. This design enables POCO to condition individual neuron predictions on past global population activity while scaling efficiently to large neural populations through the PerceiverIO architecture's cross-attention mechanism. POCO consistently outperforms every other model in our single session benchmark.

### 2.6 TRAINING

Data was chronologically partitioned into training (70%), validation (20%), and test (10%) sets. Input-target samples were generated using a sliding window approach with context length $C$ and prediction length $P$ (Fig. 8). All models were trained with the Adam optimizer (lr=$10^{-4}$) and early stopping (patience=3). Full training details are provided in Appendix B.

### 2.6.1 LOSS FUNCTION

Following standard practice in time-series forecasting, we use Mean Absolute Error (MAE) loss. The POCO paper (Duan et al., 2025) uses Mean Squared Error (MSE) loss averaged over the $P$ prediction steps. We opt for MAE as it is more robust to outliers in calcium imaging data.

$$f\left(\mathbf{x}_{t-C:t}^{(j)}, j\right) = \hat{\mathbf{x}}_{t:t+P}^{(j)}, \quad L(f) = E_{j,t}\left[\frac{1}{PN_j}\left\|\hat{\mathbf{x}}_{t:t+P}^{(j)} - \mathbf{x}_{t:t+P}^{(j)}\right\|_1\right]$$

Note that we only train and evaluate on one session $j$. The loss is averaged across all dimensions: neurons ($N$), prediction time steps ($P$), and batch samples. During training, gradients are backpropagated through this fully-averaged loss, ensuring that each neuron and time step contributes equally to parameter updates regardless of the population size.

## 3 RESULTS

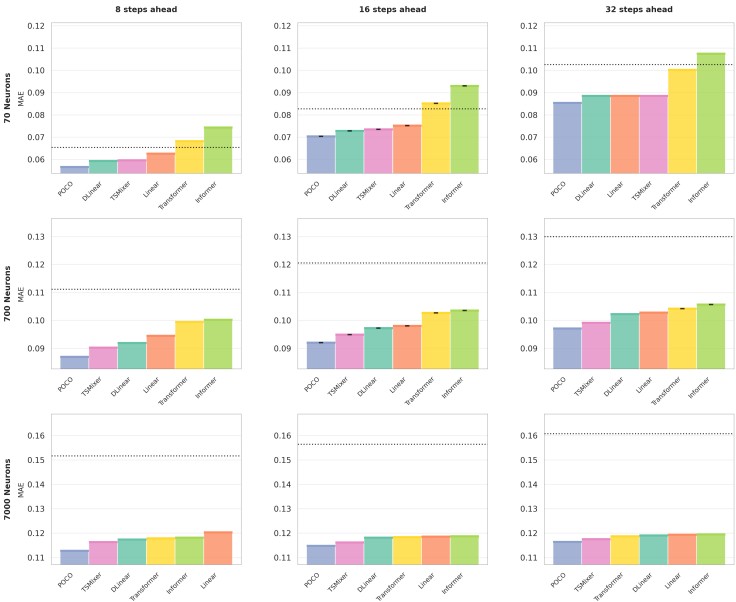

Figure 3: **Medium context window ($C = 32$) test set results.** Average MAE test set results for 70, 700, and 7000 neurons (lower is better). The prediction window length (8, 16, 32) . The dotted black line is the Naive Repeat baseline. Black error bars indicate error across 3 random seeds.

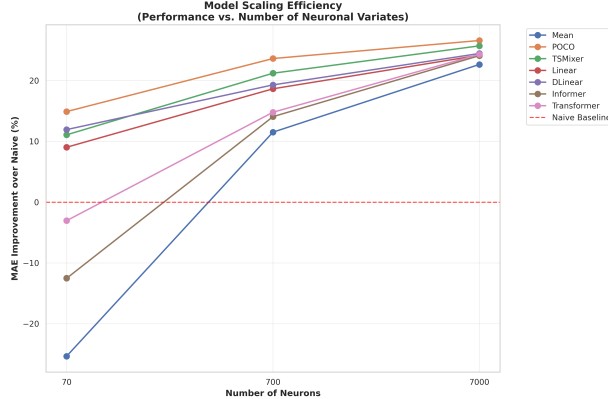

Figure 4: **All models converge to a 22-27% improvement over the Naive Repeat baseline.** Medium context window ($C = 32, P = 16$) MAE percent change over the Naive baseline provides a more intuitive insight instead of raw MAE values. We would expect multivariate models to outperform univariate ones significantly at higher neurons counts. Actual performance gains are in the single digits. This suggests multivariate models are not learning spiking patterns effectively.

## 3.1 POPULATION SCALING & CONTEXT LENGTH EFFECTS

Table 1: Grand Average Results by Population Size and Prediction Length

| Neurons | $(C, P)$ | Naive | Mean | DLinear | Linear | POCO | TSMixer | Informer | Transformer |
|---|---|---|---|---|---|---|---|---|---|
| 70 | (16,8) | 0.065 | 0.082 | 0.059 | 0.063 | **0.057** | 0.060 | 0.074 | 0.068 |
| | (48,16) | 0.083 | 0.104 | 0.073 | 0.075 | **0.070** | 0.074 | 0.093 | 0.085 |
| | (96,32) | 0.103 | 0.116 | 0.089 | 0.089 | **0.085** | 0.089 | 0.108 | 0.100 |
| 700 | (16,8) | 0.111 | 0.100 | 0.092 | 0.094 | **0.087** | 0.090 | 0.100 | 0.099 |
| | (48,16) | 0.121 | 0.107 | 0.097 | 0.098 | **0.092** | 0.095 | 0.104 | 0.103 |
| | (96,32) | 0.130 | 0.110 | 0.102 | 0.103 | **0.097** | 0.099 | 0.106 | 0.104 |
| 7000 | (16,8) | 0.152 | 0.120 | 0.117 | 0.120 | **0.113** | 0.116 | 0.118 | 0.118 |
| | (48,16) | 0.156 | 0.121 | 0.118 | 0.119 | **0.115** | 0.116 | 0.119 | 0.118 |
| | (96,32) | 0.161 | 0.121 | 0.119 | 0.119 | **0.116** | 0.118 | 0.119 | 0.119 |

Table 2: **Multivariate and univariate grand average test results on the last 10% of the session.** The vertical line after Linear denotes the beginning of multivariate models. The **best results** are highlighted in bold. All values are MAE (lower is better). Models were trained on 3 random seeds, but all values were within .05% error.

**POCO achieves the lowest MAE across all conditions**, demonstrating the effectiveness of its population conditioning mechanism. At 7000 neurons, POCO (MAE: 0.113-0.116) outperforms TSMixer (0.116-0.118) and univariate models, with this advantage most pronounced at larger population sizes where cross-neuron dependencies become more informative.

**Population scaling exhibits diminishing returns.** The 10-fold increase from 70 to 700 neurons yields a 108% improvement over the Naive baseline, while the subsequent 10-fold increase to 7000 neurons provides only 22% additional improvement. This suggests a threshold beyond which additional neurons encode increasingly redundant information for forecasting.

**Multivariate models demonstrate clear advantages only at higher neuron counts.** At 70 neurons, univariate Linear (MAE: 0.063-0.089) performs competitively with POCO (0.057-0.085) and TSMixer (0.060-0.089). However, at 7000 neurons, multivariate models establish superiority: POCO achieves 0.113-0.116 while Linear degrades to 0.119-0.120. This crossover suggests multivariate architectures require substantial population sizes to leverage cross-neuron dependencies effectively.

**Model performance converges at high neuron counts.** The MAE range narrows from 0.057-0.108 (70 neurons) to 0.113-0.121 (7000 neurons), indicating that architectural differences matter less as population size increases. Context length effects also interact with population size: at 70 neurons, longer context provides minimal benefit, while at 7000 neurons it becomes a critical factor for performance.

## 3.2 OUT-OF-DISTRIBUTION CORTICAL STATE GENERALIZATION

Table 3: **All models perform worse on the passive viewing period** ($C$=48, $P$=16, $N$=11,393). All neurons in the active period were used to train a single set of models. The same checkpoint was then evaluated on two size-matched test sets (3,765 timesteps each): a held-out active segment and the passive viewing period. This controlled comparison isolates the effect of cortical state. Prediction Score is the MSE improvement over the Naive baseline, following (Duan et al., 2025). Lower MAE is better, higher Prediction Score is better.

| Dataset | Naive MAE | DLinear MAE | DLinear Score | Linear MAE | Linear Score | POCO MAE | POCO Score | TSMixer MAE | TSMixer Score | Informer MAE | Informer Score | Transformer MAE | Transformer Score |
|---|---|---|---|---|---|---|---|---|---|---|---|---|---|
| Active Task | 0.152 | 0.115 | 0.401 | 0.116 | 0.398 | **0.112** | **0.428** | 0.113 | 0.421 | 0.116 | 0.373 | 0.116 | 0.375 |
| Passive Task | 0.163 | 0.125 | 0.390 | 0.126 | 0.381 | **0.123** | 0.394 | 0.123 | **0.401** | 0.133 | 0.281 | 0.132 | 0.289 |

**All models degrade on the passive cortical state.** A single set of models was trained on the active period and evaluated on two size-matched test sets: a held-out active segment and the passive viewing period. Every architecture exhibited increased MAE on the passive test set, confirming that the degradation is attributable to cortical state rather than any artefact of the evaluation procedure. Prediction scores decreased universally, with Informer and Transformer suffering the largest drops (0.373→0.289 and 0.375→0.281, respectively), while POCO and TSMixer proved more robust to the state transition. Neural activity statistics differed between states: the active period exhibited mean=0.316, std=0.187, while the passive period showed elevated variance (mean=0.343, std=0.216), consistent with the distinct synchronized dynamics of the disengaged cortical state (Harris & Thiele, 2011).

**Models might be learning GCaMP6s decay patterns rather than "spiking" dynamics.** Many neurons exhibit sparser activity throughout the session. After investigating the 100 lowest mean activity neurons, we found that all models were incapable of predicting instantaneous changes in fluorescence. This is most likely a result of an unsuitable loss function which prioritizes accuracy over the entire neuron's trace rather than the event-like nature of spiking.

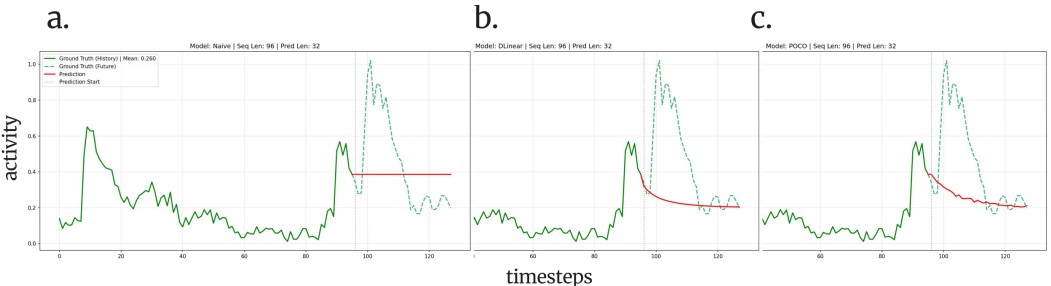

Figure 5: **Sample test results for neuron #40, ($C$=96, $P$=32) from: Naive baseline (a), DLinear (b) and POCO (c)**. Historical values (solid green) are used to predict the future P timesteps (red), MAE loss is computed between the future ground truth (dotted green) and the predicted sequence. On sparser neurons, models are unable to capture future spiking dynamics, likely a result of the current loss formulation.

## 4 CONCLUSION

We presented a systematic evaluation of neural population forecasting on large-scale calcium imaging data from a behaviorally-engaged mouse. Our experiments reveal diminishing returns with population scaling, universal degradation when generalizing to an unseen cortical state, and surprising competitiveness of univariate models.

Although our data is aligned with trial stimuli, we focus on forecasting purely from past activity, supported by ZAPBench's finding that stimulus covariates did not improve prediction ((Lueckmann et al., 2025)). Notably, models appear to learn the slow decay dynamics of GCaMP6s rather than predicting event onsets, standard forecasting losses may be unsuitable for the event-like nature of calcium traces. Candidate alternatives include transient-weighted losses that upweight timesteps with large temporal derivatives, or derivative-matching terms that penalize errors in the rate of change of the predicted signal. It might be useful to assess whether predictions on ZAPBench or other calcium imaging datasets exhibit similar decay characteristics. Moreover, it is difficult to interpret benchmark results when all models share the same failure mode.

Our benchmark establishes a foundation for tracking methodological progress as the IBL and other datasets expand. The gap between current performance and optimal prediction represents an opportunity for fundamentally new approaches to learning from population activity.

MEANINGFULNESS STATEMENT

A meaningful representation of life captures interpretable, generalizable principles underlying biological systems. Neural activity uniquely integrates genetic, morphological, and environmental

influences into population dynamics that drive perception and behavior. The brain continuously predicts future states to guide interaction with its environment; making neural forecasting a natural lens for understanding this computational process. Our work benchmarks how accurately we can predict future population dynamics from past activity, the ultimate goal being a representation of future brain dynamics.

ACKNOWLEDGMENTS

The authors thank the International Brain Laboratory for providing access to the calcium imaging dataset used in this work.

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

## A EXPERIMENTAL DETAILS

### A.1 REPRODUCIBILITY

Note that the IBL dataset is currently unpublished, but will be available to the community in the future following standard IBL practices. Code is available at: `https://github.com/the-rich-piana/dlinear-brain-forecasting`

### A.2 COMPUTE

All models were trained on a single NVIDIA RTX 3090 with 24gb of VRAM. Computational requirements are generally low to reproduce these results.

## B TRAINING PROTOCOL

### B.1 SAMPLE GENERATION

The preprocessed neural activity matrix was chronologically partitioned and input-target samples were generated using a sliding window approach, as is standard in forecasting (Fig. 8). Each sample consists of an input context window of length $C$ (context length) and a corresponding target window of length $P$. The partitioning strategy differs between experiments, as described below.

### B.1.1 POPULATION SCALING EXPERIMENTS

For the population scaling and prediction horizon experiments, subsets of 70, 700, and 7000 neurons were selected from the full population. Neurons are ordered by cortical region (as produced by Suite2p processing per field-of-view), and subsets were selected by taking the first $N$ neurons from this ordering. This deterministic selection means smaller subsets are regionally concentrated—the 70-neuron subset falls entirely within a single cortical region—while larger subsets span progressively more regions. We opted for this simple, reproducible approach rather than random or stratified sampling; future work could examine whether regionally balanced subsets alter the scaling dynamics. Data from the active task portion of the session was chronologically partitioned into training (70%), validation (20%), and test (10%) sets to prevent temporal data leakage.

### B.1.2 OUT-OF-DISTRIBUTION CORTICAL STATE EXPERIMENT

The cortical state generalization experiment is entirely separate from the population scaling experiments and uses all 11,393 neurons. The active period (18,309 timesteps, ~62.4 min) was partitioned into train (timesteps 0–11,312; ~38 min), validation (11,312–14,544; 3,232 steps, ~11 min), and active test (14,544–18,309; 3,765 steps, ~13 min). The passive period (18,309–22,074; 3,765 steps, ~13 min) serves as the out-of-distribution test set. The two test sets are size-matched (3,765 timesteps each) to ensure metric comparability. A single set of models was trained once on this split; the same checkpoint was then evaluated independently on the active and passive test sets.

### B.2 TRAINING PARAMETERS

All models were trained using the Adam optimizer with an initial learning rate of $1 \times 10^{-4}$. A learning rate scheduler was employed to reduce the learning rate if the validation loss plateaued. We used a batch size of 8. To prevent overfitting and reduce training time, we utilized an early stopping mechanism with a patience of 3 epochs; training was halted if the validation loss did not improve for three consecutive epochs. Models were trained for a maximum of 10 epochs. All models were implemented in PyTorch.

## C RECORDING DETAILS

This dataset is unpublished and will be part of a separate publication in the future. Nonetheless, we provide the recording details that were available to us at the time of writing.

### C.1 EXPERIMENTAL SETUP AND MESOSCOPE IMAGING

Data was collected from a transgenic mouse (P063) expressing GCaMP6s in excitatory neurons (CamKII-tta x tetO-GCaMP6s) performing the IBL standardized decision-making task. The IBL task requires mice to classify visual stimuli (static gratings appearing either to the left or right of the visual field) and respond by turning a wheel in the corresponding direction to receive water rewards. We experimented with 12 other sessions, but ultimately selected one from mouse P063, since it was free of quality control issues.

### C.2 SURGICAL PREPARATION AND WINDOW IMPLANTATION

Prior to data collection, the mouse underwent surgical implantation of a chronic imaging window. A 4-5mm diameter glass window was implanted and centered approximately over the right visual area (VISa), positioned at stereotaxic coordinates ~2.5 ML and ~-2.7 AP relative to bregma. This window placement provided optical access to multiple cortical regions across the posterior cortex.

### C.3 2-PHOTON CALCIUM IMAGING PARAMETERS

Neural activity was recorded using 2-photon calcium imaging across 8 fields-of-view (FOVs) within the chronic window. Each FOV covered $585 \times 585$ $\mu$m with $512 \times 512$ pixel resolution in a single imaging plane. The recording captured 22,074 time points at a sampling rate of 4.9 Hz, yielding 4,504 seconds of continuous neural data. The imaging session protocol consisted of two phases: **(1)** 61.1 minutes of active task behavior (biasedChoiceWorld task) containing 521 trials, followed by **(2)** 14.1 minutes of passive imaging protocol with zebra noise stimuli, locked wheel, and no water rewards. ROI extraction and neural signal processing was performed using Suite2p, a standard calcium imaging analysis pipeline that handles motion correction, cell segmentation, and signal deconvolution.

### C.4 NEURAL POPULATION AND REGIONAL COVERAGE

The recording captured activity from 11,393 neurons distributed across 11 cortical regions.

| Cortical Region | Neuron Count | Percentage |
|---|---|---|
| Primary visual cortex (VISp1) | 2,856 | 25.1% |
| Primary somatosensory cortex, barrel field (SSp-bfd1) | 2,439 | 21.4% |
| Rostrolateral visual area (VISrl1) | 2,414 | 21.2% |
| Primary somatosensory cortex, trunk (SSp-tr1) | 1,471 | 12.9% |
| Retrosplenial area, lateral agranular part (RSPagl1) | 788 | 6.9% |
| Anterior visual area (VISa1) | 774 | 6.8% |
| Retrosplenial area, dorsal part (RSPd1) | 259 | 2.3% |
| Anterolateral visual area (VISal1) | 249 | 2.2% |
| Anteromedial visual area (VISam1) | 70 | 0.6% |
| Primary somatosensory cortex, upper limb (SSp-ul1) | 51 | 0.4% |
| Primary somatosensory cortex, lower limb (SSp-ll1) | 22 | 0.2% |

Table 4: Regional distribution of recorded neurons across cortical areas. Total neuron count: 11,393.

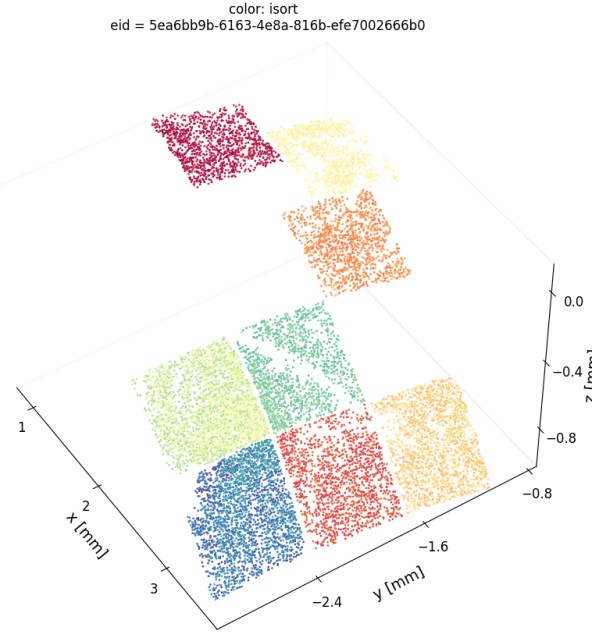

Figure 6: 3D visualization of recorded neuron locations colored by isort index. Each point represents a single neuron's spatial coordinates (x, y, z) in millimeters within the imaging field of view.

## C.5 DATA PROCESSING AND QUALITY CONTROL

Raw imaging data underwent standard processing using the IBL data pipeline with modifications for 2-photon mesoscope data synchronization. Quality control of raw TIFF files was performed using semi-automated algorithms, followed by metadata extraction from TIFF headers combined with experimental parameters. Allen Common Coordinate Framework (CCF) registration was applied to infer ML-AP-DV location of each pixel. Motion correction, segmentation, and deconvolution were performed using Suite2p, after which neural traces were synchronized with behavioral task events and registered to the IBL database in ALF (ALyx File) format.

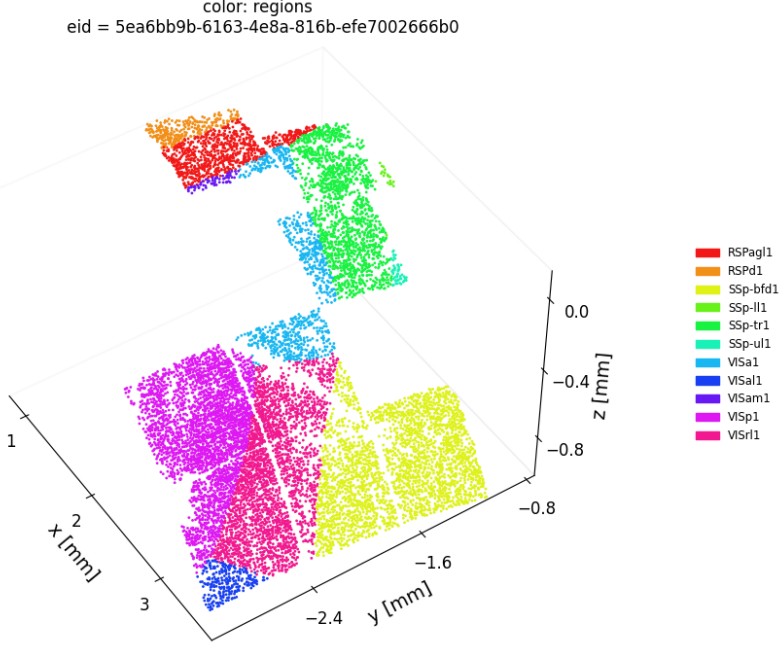

Figure 7: 3D visualization of ROIs colored by cortical region. Each point represents a single neuron's spatial coordinates, with colors indicating distinct cortical areas.

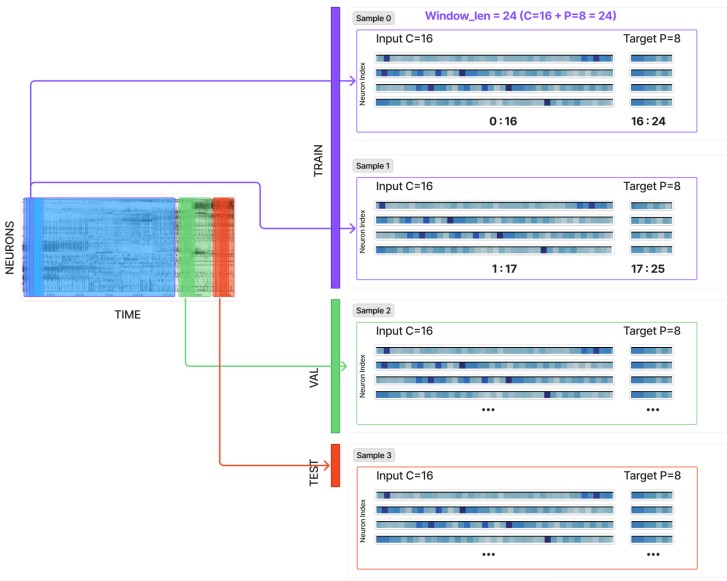

Figure 8: Sample generation and partitioning of the activity matrix for timeseries forecasting. The neural population data matrix is chronologically partitioned into training, validation, and test sets. Sliding window sampling generates input-target pairs with context length C and prediction length P, creating overlapping samples that slide one timestep forward across the temporal dimension.

