# OpenReview forum: "A Forecasting Benchmark of Large-Scale Neural Populations during Task-Driven Behavior"
_ICLR.cc/2026/Workshop/LMRL — ICLR 2026 Workshop LMRL Poster_

### Official Review · Reviewer_1Xjf · 2026-02-27
**Benchmark paper on neural activity prediction - needs some expansion/clarification.**

**Rating:** 7
**Confidence:** 4

**Review:**

Summary
This paper benchmarks neural population forecasting on a large-scale
IBL calcium imaging dataset (11,393 neurons - though only 7,000 are
used in the largest experiments), evaluating models across population
scales, prediction horizons, and cortical states. The findings are
interesting and worth communicating to the community, though there are
some concerns.

Major Concern: Confounding between state and train/validation/test
(figure 2)
suggests that training and validation are drawn entirely from the
active portion of the session, while the test set is the passive
portion. Is this the source of the performance drop in active
vs. passive prediction accuracy?  It seems like an out-of-sample
prediction problem. To isolate the cortical state effect, the authors
should include a held-out active test segment as a baseline
comparison. As currently designed, the accuracy drop may have nothing
to do with the particular cortical state per se.

Minor Conern: Is spatial structure useful in prediction or evaluation?
The dataset spans 11 cortical regions with known spatial coordinates,
yet none of the models incorporate this structure, and it doesn't seem
to be used for evaluation (e.g., does attention focus on nearby
neurons?). The authors should discuss how spatial information could
inform analyses or be used as priors, if possible.

Minor Concern: How were subpopulations chosen?
Relatedly, how were the 70/700/7000 neuron subsets selected? Random,
regionally stratified, or by activity level? This choice could
meaningfully affect the multivariate scaling results.

Minor Concer: Loss function choice
The authors note that models learndecay dynamics rather than
spike-like events but do not propose an alternative loss. A concrete
discussion of candidate approaches (e.g., transient-weighted losses or
spike train metrics) would make this observation much more useful.

Minor Concern: Model Selection
The model suite may be broader than the conclusions require.  It may
be more useful to reduce down to 3 or 4 models (simplest and most
complex, plus 1 or 2 more).  As it is, there's more attention


Overall
The dataset is interesting and the experimental setup is carefully
described (though these are from other forthcoming papers?). The
cortical state confound is the most pressing issue. Clarifying neuron
subset selection and providing guidance on loss function design would
substantially improve the paper's utility as a benchmark resource.

---

### Meta-Review · Area_Chair_34oc · 2026-02-28

**Recommendation:** Accept (Poster)
**Confidence:** 4

**Metareview:**

Accept.

---

### Decision · Program_Chairs · 2026-03-02

**Decision:**

Accept (Poster)

**Comment:**

Please see the meta-review.